# Whey Protein Isolate-*Mesona chinensis* Polysaccharide Conjugate: Characterization and Its Applications in O/W Emulsions

**DOI:** 10.3390/foods12051068

**Published:** 2023-03-02

**Authors:** Meixiang Yao, Xin Qi, Jiahui Zhang, Chengyuan Wang, Jianhua Xie

**Affiliations:** 1Jiangzhong Dietary Therapy Technology Co., Ltd., Jiujiang 332020, China; 2State Key Laboratory of Food Science and Technology, Nanchang University, Nanchang 330047, China; 3International Institute of Food Innovation, Nanchang University, Nanchang 330200, China

**Keywords:** *Mesona chinensis* polysaccharide, whey protein isolate, conjugate, emulsion

## Abstract

*Mesona chinensis* polysaccharide (MCP), a common thickener, stabilizer and gelling agent in food and pharmaceuticals, also has antioxidant, immunomodulatory and hypoglycemic properties. Whey protein isolate (WPI)-MCP conjugate was prepared and used as a stabilizer for O/W emulsion in this study. FT-IR and surface hydrophobicity results showed there could exist interactions between -COO- in MCP and -NH^3+^ in WPI, and hydrogen bonding may be involved in the covalent binding process. The red-shifted peaks in the FT-IR spectra suggested the formation of WPI-MCP conjugate, and MCP may be bound to the hydrophobic area of WPI with decreasing surface hydrophobicity. According to chemical bond measurement, hydrophobic interaction, hydrogen bond and disulfide bond played the main role in the formation process of WPI-MCP conjugate. According to morphological analysis, the O/W emulsion formed by WPI-MCP had a larger size than the emulsion formed by WPI. The conjugation of MCP with WPI improved the apparent viscosity and gel structure of emulsions, which was concentration-dependent. The oxidative stability of the WPI-MCP emulsion was higher than that of the WPI emulsion. However, the protection effect of WPI-MCP emulsion on β-carotene still needs to be further improved.

## 1. Introduction

Compared with conventional emulsion, Pickering emulsion has the advantages of enhanced encapsulation capability and resisting coalescence, phase separation, and Ostwald ripening [1,2,3]. Therefore, in recent years, Pickering emulsion is often used as carriers for the delivery of bioactive substances, including lutein, curcumin, resveratrol and so on. Generally, Pickering emulsion is stabilized by organic solid particles like protein, phospholipid, polysaccharide, and inorganic solid particles like SiO_2_ and CaCO_3_. Food grade organic particles have won people’s favor because of their safe, green, effective features [3]. In addition, it is difficult to prepare superior emulsions with a single stabilizer and thus protein-polysaccharide, protein-phospholipid, and protein-protein composite nanoparticles have attracted widespread attention. Xu et al. [4] prepared a WPI-chitosan complex stabilized emulsion for controlled and sustainable release of α-tocopherol. Liu et al. [5] stabilized O/W Pickering emulsion with WPI glycated with glucose, lactose, and maltodextrin. They found that glycation changed the surface hydrophobicity of WPI, improved protein adsorption, and formed a more stable emulsion. Previous studies also showed that polysaccharide improved the performance of protein-based emulsion systems, such as increased encapsulation efficiency (EE), improved redispersibility after drying and bioaccessibility [6,7]. Therefore, protein-polysaccharide composite nanoparticles are promising stabilizers in emulsions. 

MCP is an acid heteropolysaccharide extracted from *Mesona chinensis* Benth (Figure 1b), a kind of medicinal and edible plant of the Labiatae family [8]. It is composed of xylose and galacturonic acid [9]. MCP has good rheological properties and gelling behavior, and a great influence on the textural, rheological, and digestibility properties of starch. Some studies suggested that it could be used to prepare starch-based food packaging materials, nanoparticles that deliver bioactive substances, and self-supporting hydrogels with the desired texture and gelling properties [10]. In addition, MCP possesses various bioactivities, including antioxidant effect against DPPH radical and ABTS radical cation, hepatoprotective and immunoregulatory effects [11]. However, study on MCP-based emulsions is scarce.

β-carotene, a natural lipophilic compound, is widely found in many fruits and vegetables, such as pawpaw, carrot, sea buckthorn, and smoke stove. Owing to its excellent coloring effect and biocompatibility, β-carotene is allowed to be used as an additive in the food industry (e.g., fruit juice, candies, jam, flavored fermented milk, etc.) for coloring purposes [12]. β-carotene contains eight isoprene structures on the main chain and two β-viologen ring structures at the end (Figure 1a), which gives it favorable antioxidant activity. β-carotene is often used to prevent oils from oxidizing and prepare food packaging material as an excellent natural antioxidant [13,14]. In addition to these in vitro applications, β-carotene can perform many important biological functions in the human body as a vitamin A precursor [15]. β-carotene possesses antioxidation, anti-inflammatory and anti-cancer activities [12,16]. Furthermore, it can improve intestinal dysfunctions, control defects in vision, modulate atherosclerotic cardiovascular disease, and reduce the symptoms of Alzheimer’s disease [17,18,19,20]. However, the unsaturated double bonds in the β-carotene molecule structure cause poor storage stability and thus limits its wide application in the food field [21]. 

In this study, the MCP was covalently combined with WPI, a kind of protein with good gelation, emulsification, and polysaccharide binding properties, to prepare WPI-MCP conjugate. The WPI-MCP conjugate was characterized by FT-IR, chemical bonds, and surface hydrophobicity measurements. Then, WPI-MCP conjugate was used to prepare O/W emulsion, and its protective effect on β-carotene during storage was investigated. Knowledge obtained from our work will contribute to the development of polysaccharide-protein conjugate stabilized emulsions and provide information for better stability of β-carotene.

## 2. Materials and Methods

### 2.1. Materials

MCP (35.62% total sugar, 37.14% of uronic acid, 14.33% protein, and the molecular weight was 325 kDa) was extracted according to Lin et al. [22]. WPI was purchased from Hilmar Ingredients Corporation (Hilmar, CA, USA), and β-carotene was obtained from Aladdin Biochemical Technology Co., Ltd. (Shanghai, China). Corn oil was obtained from the local market (Nanchang, China). 1-anilino-8-naphthalensulfonate (ANS) was purchased from Shanghai Yuanye Bio-Technology Co., Ltd. (Shanghai, China). The water used in this work was ultrapure water.

### 2.2. Preparation and Characterization of WPI-MCP Conjugate

#### 2.2.1. Preparation

MCP and WPI were dissolved in phosphate buffer (PBS) (10 mM, pH 7.0) with magnetic stirring at 25 °C for 120 min respectively. The MCP and WPI stock solutions were placed at 4 °C overnight for hydration. After that, MCP stock solutions were mixed with WPI stock solutions and named WPI (2% WPI, *w/v*), WPI-MCP_0.05_ (2% WPI + 0.05% MCP, *w/v*), WPI-MCP_0.1_ (2% WPI + 0.1% MCP, *w/v*), WPI-MCP_0.2_ (2% WPI + 0.2% MCP, *w/v*), WPI-MCP_0.3_ (2% WPI + 0.3% MCP, *w/v*), respectively, and then heated at 95 °C for 30 min. The mixtures were placed in ice to cool immediately, followed by homogenization at 13,000 r/min for 5 min to prepare the composite particles (WPI-MCP conjugate).

#### 2.2.2. FT-IR

Samples were lyophilized using a FreeZone 2.5 freeze dryer (Labconco, Kansas City, MO, USA) and taken at approximately 1:100 with potassium bromide, and the infrared spectra were obtained on a Nicolet 5700 Fourier transform infrared spectrophotometer (Nicolet, Madison, WI, USA) using potassium bromide as a blank background control. The wavelength range was 4000–500 cm^−1^ with 64 scans and 4 cm^−1^ resolution. Each sample was measured three times.

#### 2.2.3. Chemical Bonds Measurement

The chemical bonds that existed in WPI-MCP conjugate containing 2% WPI and 0.2% MCP (*w/v*) were evaluated based on Deng’s [23] method with slight modification. 0.2% MCP concentration (*w/v*) was chosen because the WPI-MCP_0.2_ emulsion had the smallest particle size (not shown in this work). Four solvents were prepared: solvent 1 (S1) was 0.6 M sodium chloride, solvent 2 (S2) was 0.6 M sodium chloride + 1.5 M urea, solvent 3 (S3) was 0.6 M sodium chloride + 8 M urea, and solvent 4 (S4) was 0.6 M sodium chloride + 8 M urea +0.5 M β-mercaptoethanol. 500 mg lyophilized samples were added to 5 mL S1 and then mixed by vortexing for 120 s. The mixtures were placed at 25 °C for 20 min, followed by centrifugation at 10,000× *g* for 20 min. The precipitates were mixed with 5 mL S2/S3/S4 with the same procedures. The WPI solubility, expressed as the percentage of protein content (obtained by Bradford method) in the supernatant relative to the total protein, in S1, S2, S3, and S4 was used to evaluate the ionic bond, hydrogen bond, hydrophobic interaction, and disulfide bond, respectively.

#### 2.2.4. Surface Hydrophobicity

Surface hydrophobicity was determined using Dong’s method [24]. ANS working solution was obtained by dissolving ANS in PBS (10 mM, pH 7.0) buffer. The WPI and WPI-MCP systems were diluted with the same PBS to a protein concentration from 0.1 to 0.5 mg/mL at 0.1 mg/mL intervals. Surface hydrophobicity of MCP was obtained at 0.1, 0.2, 0.3, 0.4, and 0.5 mg/mL MCP concentration. The dilution was added to 8 mM ANS solution at a ratio of 200:1, mixed well and placed in the dark for 15 min before the measurement of fluorescence intensity on a microplate reader (Molecular Devices, Sunnyvale, CA, USA). The excitation, emission wavelengths, and slit width were 390, 470, and 10 nm respectively. The initial slope of the fluorescence intensity versus WPI concentration corresponded to the surface hydrophobicity. 

### 2.3. Preparation and Characterization of WPI-MCP Emulsion

#### 2.3.1. Preparation

WPI-MCP solutions containing 2% (*w/v*) WPI and 0%, 0.05%, 0.1%, 0.2%, and 0.3% (*w/v*) MCP were mixed with corn oil at a ratio of 9:1 (*v/v*). The mixtures were homogenized with a high-speed shear emulsifier at 13,000 r/min for 5 min, and then the emulsion was obtained by a high-pressure microjet circulating three times at 120 MPa.

#### 2.3.2. Microstructure Analysis

The morphology of the Pickering emulsion was characterized by an Olympus CKX53 microscope (Olympus Co., Ltd., Tokyo, Japan). A 10-fold dilution of the emulsion was placed on the slide without a coverslip to avoid deformation of the droplets to observe the microstructure. In addition, a BX 53 fluorescence microscope (Olympus Co., Ltd., Tokyo, Japan) was employed for observing the emulsion interfacial structure. Nile Red (0.1%, *w/v*) and Nile Blue A dye (0.1%, *w/v*) were used for staining, which were excited by a 488 nm argon laser and a 633 nm helium-neon (He-Ne) laser, respectively. 

#### 2.3.3. Rheological Properties

Rheological properties of emulsions were obtained on the DHR-2 rheometer (TA Instruments Inc., New Castle, DE, USA) using a 40 mm diameter parallel plate at a 0.5 mm gap at 25 °C after 12 h of resting. For steady rheological determination, the relationship between apparent viscosity and the shear rate was recorded at the shear rate ranging from 0.1 to 100 s^−1^. For dynamic viscoelasticity properties, the changes of storage modulus (G′) and loss modulus (G″) were recorded at 0.1–10 rad/s frequency range.

### 2.4. Preparation and Characterization of β-Carotene-Loaded WPI-MCP Emulsion

#### 2.4.1. Preparation

β-carotene was dissolved in corn oil at 1 mg/mL under ultrasonic processing for 30 min. The emulsions were prepared according to Section 2.3.1. In the WPI-MCP β-carotene emulsion, the concentration of β-carotene was 0.1 mg/mL, and the WPI-MCP solution containing 0.2% (*w/v*) MCP was chosen to prepare β-carotene emulsion because the particle size of the emulsion stabilized by WPI-MCP_0.2_ was the smallest (not shown in this work). Emulsion without MCP as a control group.

#### 2.4.2. Particle Size and Zeta Potential Determinations

100-fold dilution of the emulsion was measured for particle size and zeta potential using a Zetasizer Nano ZS90 particle size analyzer (Malvern Inc., Malvern, UK). The determinations were preformed in triplicate at 25 °C.

#### 2.4.3. EE

EE of β-carotene was evaluated according to Zhang’s [10] method with adaptations. The β-carotene entrapped within the emulsion was extracted by anhydrous ethanol-n-hexane (1:2, *v/v*) solution 3 times. The pooled n-hexane phase was measured on a microplate reader (Molecular Devices, Sunnyvale, CA, USA) at 450 nm. The amount of β-carotene in the sample was calculated by the standard curve. EE was obtained by the following equation:(1)$$\mathrm{EE}\;\left(\%\right)=\frac{\mathrm{Entrapped}\;\beta -\mathrm{carotene}}{\mathrm{Total}\;\mathrm{mass}\;\mathrm{of}\;\mathrm{input}\;\beta -\mathrm{carotene}}$$

#### 2.4.4. Oxidative Stability Assessment

The peroxide value (POV) was analyzed by measuring the content of hydrogen peroxide in the primary lipid of the emulsion based on Yuan’s [25] method with slight modification. The emulsion was mixed with isopropanol-isooctane solution (1:2, *v/v*) at a ratio of 1:5. Then, 200 μL of supernatant were taken after centrifugation at 3500× *g* for 2 min and mixed with 20 μL 3.94 mol/L thiocyanate and 20 μL of Fe^2+^, and then fixed to 5 mL with butanol-methanol solution (1:2, *v/v*). The mixtures were kept away from light for 20 min, followed by a record on a microplate reader (Molecular Devices, Sunnyvale, CA, USA) at 510 nm. The concentration of peroxides was calculated by a standard curve prepared with Fe^3+^. 

Thiobarbituric acid reactive substances (TBARS) were analyzed according to Chen’s method [26]. Emulsions were mixed with trichloroacetic acid (10%, *w/v*) and thiobarbituric acid solution (1%, *w/v*) at a ratio of 3:5:2, followed by heat treatment at 100 °C for 30 min. The emulsions were placed in ice for rapid cooling, followed by centrifugation at 4500 r/min for 20 min. The supernatant was collected and measured on a microplate reader (Molecular Devices, Sunnyvale, CA, USA) at 532 nm. Different concentrations of 1,1,3,3-tetraethoxypropane (0, 1.25, 2.5, 5, 10, 20 µM) were used to calculate the TBARS value. The samples were placed at 45 °C and the POV and TBARS values were analyzed at 0, 7, 14, 21, and 28 days. 

### 2.5. Chemical Stability Analysis

The WPI β-carotene emulsion and WPI-MCP β-carotene emulsion were put in centrifuge tubes respectively and then stored at 4 °C and 25 °C away from light. The β-carotene retention, expressed as C_t_/C_0_, where C_0_ and C_t_ were the β-carotene content at the 0 and t days storage, respectively, was measured at 0, 7, 14, 21, and 28 days by the same method with Section 2.4.3. 

### 2.6. Statistical Analysis

Data were analyzed by one-way analysis of variance (ANOVA) using SPSS 26.0 software (IBM, Chicago, IL, USA) and reported as mean ± SD. There was significant difference when the value of *p* < 0.05.

## 3. Results and Discussion

### 3.1. FT-IR

Infrared spectroscopy can be used to study the structure and chemical bonding of compounds [27]. Figure 2a showed the FT-IR spectra of WPI, MCP, and WPI-MCP conjugates at the 4000–500 cm^−1^ wavenumber range. For MCP, the peaks at 3349 cm^−1^ and 2936 cm^−1^ corresponded to the stretching vibrations of O-H and C-H, respectively [22]. 1608 cm^−1^ could be caused by the carbonyl C = O vibrations in uronic acid. For WPI, the peak at 3292 cm^−1^ represented the stretching vibrations of O-H. Peaks at 1645 cm^−1^ (amide I) and 1537 cm^−1^ (amide II) are attributed to C = O stretching vibrations and C-N stretching vibrations in combination with N-H bending, respectively. After interaction with MCP, the peak of WPI at 3292 cm^−1^ shifted to 3294 cm^−1^ and became wider, indicating the binding of MCP with WPI and the intermolecular and/or intramolecular hydrogen bonds in WPI-MCP conjugate increased. Tirgarian et al. [28] also reported that conjugation of soy protein isolate (SPI) and sodium caseinate with polysaccharides including *Alyssum homolocarpum* seed gum and kappa-carrageenan induced red shift of the peaks of these two proteins in 3200–3500 cm^−1^. Moreover, the peak in the amide I band at 1645 cm^−1^ shifted to 1647 cm^−1^ after conjugation, which may be caused by the interaction between -COO- from MCP and -NH^3+^ from WPI. The peaks at 1537, 1450, and 1394 cm^−1^ shifted to 1541, 1456, and 1398 cm^−1^, respectively. Chen et al. [29] showed that conjugation resulted in a red shift, and the stronger the conjugate effect, the stronger the red shift. From the above results, it can be seen that these peaks in amide I, amide II, and 3200–3500 cm^−1^ underwent varying degrees of redshift, which supported the formation of WPI-MCP conjugate. Additionally, the spectra of WPI-MCP conjugates with different MCP concentrations were similar. 

### 3.2. Surface Hydrophobicity

Surface hydrophobicity has been used to predict and evaluate changes in the surface properties of the protein. The hydrophobic groups on the protein surface play a key role in maintaining stable protein conformation [30,31]. ANS is an effective tool to measure surface hydrophobicity, because ANS can bind to hydrophobic amino acids on the protein surface [32]. As shown in Table 1, the surface hydrophobicity values of WPI and MCP were 29.57 and 3.07, respectively, showing that MCP processed lower surface hydrophobicity, which may be due to the presence of a large number of hydroxy groups in the MCP. After cross-linking of WPI with MCP, WPI surface hydrophobicity significantly decreased with increasing concentration of MCP. The decrease of surface hydrophobicity started to slow down at 0.2% (*w/v*) MCP concentration. At this point, the value of surface hydrophobicity was reduced by 69.33%. The addition of polysaccharides introduced hydroxyl groups, which could decrease the surface hydrophobicity of the whole system. Another reason could be that MCP may be bound to the hydrophobic area of WPI and thus reduce the binding sites of ANS on WPI. Moreover, the formation of WPI-MCP conjugate with larger molecular weight may increase the steric hindrance preventing ANS adsorption. Hu et al. [33] reported that the surface hydrophobicity of SPI-*Pleurotus eryngii* polysaccharide conjugates was lower than that of WPI. Huang et al. [34] also found the surface hydrophobicity of WPI decreased after conjugation with genipin-crosslinked alkaline soluble polysaccharides. They thought one of the reasons could be that the larger molecular weight after crosslinking increased steric hindrance for adsorption by ANS.

### 3.3. Chemical Bonds Measurement

The intermolecular force was investigated by breaking the different forces involved in protein-protein and protein-polysaccharide molecules with different solvents. The intermolecular forces that appeared in WPI and WPI-MCP solutions were evaluated in our study. Figure 2b indicated there were fewer ionic bonds in the WPI-MCP system. MCP as acidic polysaccharides had negative charges, and WPI was also negatively charged at pH 7 because its isoelectric point was pH 4.5 [35]. Therefore, it is difficult for WPI and MCP to interact with each other through ionic bonds. In addition, the hydrophobic interaction, hydrogen bond, and disulfide bond increased significantly (*p* < 0.05) after conjugation with MCP, especially the hydrogen bond. The reason may be that MCP possessed abundant hydroxyl groups, which could promote the formation of hydrogen bonds between hydroxyl groups in MCP and amino and carboxyl groups in WPI. Moreover, heat treatment may cause the WPI conformation to unfold, exposing internal sulfhydryl and hydrophobic groups, which could enhance hydrophobic interaction, and disulfide bond. Therefore, hydrophobic interaction, hydrogen bond and disulfide bond played the main role in the formation process of WPI-MCP conjugate.

### 3.4. Morphology

Figure 3 depicted the microscopic morphology of WPI and WPI-MCP emulsions. WPI was marked in red and the oil phase in green (Figure 3a). The oil droplets were encapsulated, and O/W emulsions were formed. The results showed that the WPI-MCP conjugates were surface active and had a tendency to adsorb to the oil-water interface. According to Figure 3b, the diameter of emulsion droplets increased with increased MCP concentration, which may be caused by the thick coating formed by the binding of protein with polysaccharide. The WPI-MCP conjugate formed a dense filling layer on the surface of the spherical oil droplets. This interfacial structure created a physical barrier to flocculation, coalescence, and Ostwald maturation of Pickering emulsion. 

### 3.5. Rheological Property

As shown in Figure 4a, the apparent viscosity of WPI-MCP emulsions containing 0%, 0.05%, 0.1%, 0.2%, and 0.3% MCP (*w*/*v*) was investigated. The viscosity of the emulsion decreased significantly when the shear rate increased, which indicated the emulsions exhibited shear thinning properties [36,37]. This phenomenon can be attributed to the gradual disruption of flocculated droplets and the alignment of droplets and polymers with the water flow [38]. When the emulsion was subjected to shear, the originally entangled macromolecules separated, the resistance to flow was reduced, and then the viscosity decreased. Similar phenomena have been previously reported [39]. When the concentration of MCP increased, apparent viscosity also increased. This was because the higher the polysaccharide concentration, the high molecular weight molecules in the emulsion were more likely to collide with each other, resulting in increased flow resistance and viscosity. Jiang et al. [40] suggested that MCP was a thickening agent, and when MCP was cross-linked with WPI, independently moving molecules were restricted, resulting in enhanced apparent viscosity and pseudoplastic properties.

The G′and G″ values at 0.1–10 rad/s oscillation frequencies were shown in Figure 4b. It was clear that G′ was higher than G″ in the oscillation frequency range for all samples, indicating the elastic gel-like structure formed [41]. Moreover, there was a tendency for G′ and G′′ to increase with increasing MCP concentration, and similar results were found by Lv et al. [42] in their study of WPI-chitosan emulsion, suggesting the Pickering emulsion gel structure was enhanced. The value of G′ increased gradually with increasing particle concentration, but to a lesser extent, indicating that they were essentially covalent “physical” crosslinks. Also, since MCP had higher viscosity at higher concentrations, increasing MCP concentration may also help to enhance the gel structure.

### 3.6. Size, Zeta Potential and EE of β-Carotene Emulsions

As shown in Table 2, the average particle size of the WPI β-carotene emulsion and WPI-MCP β-carotene emulsion were 175.5 ± 2.79 nm and 235 ± 2.03 nm. The larger average size may be due to the ability of the covalently bound WPI-MCP to form a macromolecular stabilization layer around the WPI layer. PDI values (<0.3) implied that the emulsions had a narrow particle size distribution. The WPI β-carotene emulsion showed higher zeta potential than the WPI-MCP β-carotene emulsion. This may be due to the partial adsorption of the anion on the MCP molecule onto the surface of the WPI particles. Conjugation of whey proteins with inulin has also been reported to lose WPI positive charge [43]. The absolute values of zeta potential for both emulsions were greater than 30 mV, indicating that the electrostatic repulsion present between droplets can maintain the emulsion stability and WPI-MCP β-carotene emulsion could be more stable [44]. EE of WPI β-carotene emulsion and WPI-MCP β-carotene emulsion were 86.68 ± 2.45 and 87.18 ± 0.67 respectively, showing that there was no significant difference (*p* > 0.05).

### 3.7. Oxidative Stability

In Figure 5, the WPI emulsion showed the highest POV and TBARS values during accelerated oxidation, suggesting it had the weakest oxidative stability. The reason may be that oil droplets were heavily accumulated and exposed to air, and the free radicals in the oil were oxidized by contacting with O_2_ in the air [45]. However, the POV and TBARS values of WPI-MCP emulsion decreased significantly, which may be due to the antioxidant capacity of the hydroxyl structure of MCP. Another reason could be that the covalent binding of MCP with WPI increased the thickness of the interfacial layer and then prevented contact between O_2_ in the air and free radicals in the oil, thus inhibiting the lipid oxidation reaction. Additionally, it was known from Section 3.5 that the viscosity of the emulsion increased in the presence of MCP, which could inhibit the movement of oxidizing radicals and metal ions, leading to high oxidative stability [46]. Notably, when β-carotene was added to the emulsion, the POV and TBARS values were significantly lower than that of the emulsion without β-carotene, indicating that β-carotene had strong antioxidant properties. There was no significant difference in POV and TBARS values between WPI-MCP β-carotene emulsion and WPI β-carotene emulsion for most of the time (*p* < 0.05), which could be caused by the strong antioxidant capacity of β-carotene. In the WPI β-carotene emulsion and WPI-MCP β-carotene emulsion systems, it was mainly the β-carotene that acted as an antioxidant. From the above results, it can be seen that small molecule antioxidants, and polysaccharides with antioxidant properties can improve the oxidative stability of emulsions and help to extend the shelf life of emulsions. 

### 3.8. Chemical Stability of β-Carotene

Due to the presence of a large number of unsaturated double bonds in β-carotene (Figure 1a), it is susceptible to oxidation and trans-isomerization under the action of light, heat, and oxygen. Generally, the degradation of β-carotene leads to the formation of multiplex degradation compounds, including isomers (13-cis-β-carotene, 13,15-di-cis-β-carotene, etc.), epoxides (β-carotene 5,6-epoxide, β-carotene 5,8-epoxide, etc.), apocarotenones, apocarotenals and short-chain cleavage products (β-cyclocitral, β-ionone, ionene, 5,6-epoxi-β-ionone, dihydroactinidiolide, 4-oxo-ionone, etc.) [47,48]. Therefore, the effects of different temperatures and stabilizer types on the chemical stability of β-carotene emulsions were investigated over a storage time of 28 days. As shown in Table 3, the retention of β-carotene in the samples tended to decrease during storage under all storage conditions. After one week of storage, no major degradation of β-carotene had yet occurred. During the 14 days of storage, there was no significant difference in β-carotene retention between WPI β-carotene emulsion and WPI-MCP β-carotene emulsion stored at 4 °C, while β-carotene retention in WPI-MCP β-carotene emulsion at 25 °C was higher than that in WPI β-carotene emulsion (*p* < 0.05), indicating WPI-MCP emulsion had better protection effect for β-carotene. After 28 days of storage, β-carotene in emulsion stabilized by WPI at 25 °C, WPI-MCP at 25 °C, WPI at 4 °C, and WPI-MCP at 4°C was 71.8 ± 4.42%, 73.57 ± 3.46%, 78.7 ± 14.03%, and 86.33 ± 6.44% of the initial levels, respectively. Β-carotene in WPI-MCP emulsion at 4 °C was significantly higher than that at 25 °C, indicating β-carotene was sensitive to the temperature. Moreover, there was no significant difference in β-carotene retention between WPI β-carotene emulsion and WPI-MCP β-carotene emulsion when stored at the same temperature for 28 days, while WPI-MCP β-carotene emulsion had a larger average value. These results indicated that storage temperature was an important factor in determining the stability of β-carotene. Compared to WPI emulsion, WPI-MCP emulsion had more potential to improve β-carotene stability, but the protection effect still needed to be improved. 

## 4. Conclusions

In this work, the WPI-MCP conjugates with lower surface hydrophobicity than WPI were prepared, and there were hydrophobic interactions, hydrogen bonds, and disulfide bonds in the conjugation process. The WPI and WPI-MCP emulsions were O/W emulsions. The increased MCP concentration led to increased viscosity, enhanced gel structure, and increased size of droplets. The POV and TBARS values of the WPI-MCP emulsion were lower than those of the WPI emulsion, indicating improved oxidative stability. The ability of WPI-MCP emulsion to protect β-carotene from degradation still needs to be improved. This work provided references for the development of emulsion stabilized by protein-polysaccharide conjugate and the theoretical basis for the potential application of WPI-MCP conjugate on anti-lipid oxidant in the emulsion.

## Figures and Tables

**Figure 1 foods-12-01068-f001:**
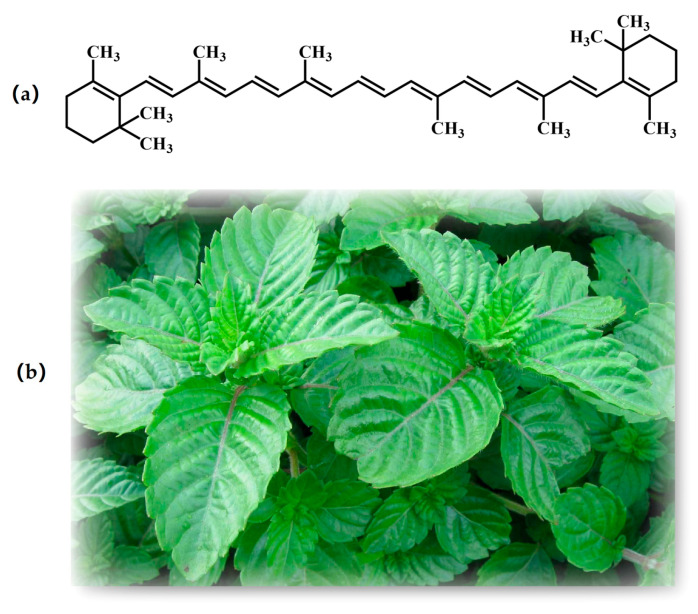
The chemical structure of β-carotene (**a**) and appearance of *Mesona chinensis* Benth (**b**). (**c**) Overall experimental plan.

**Figure 2 foods-12-01068-f002:**
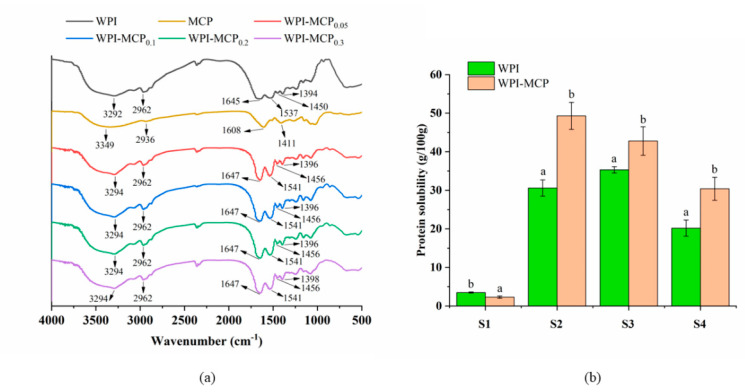
(**a**) FTIR spectra of WPI, MCP and WPI-MCP conjugates. (**b**) Intermolecular forces, inluding ionic bond (S1), hydrogen bond (S2), hydrophobic interaction (S3), and disulfide bond (S4) in WPI and WPI-MCP systems containing 2% WPI and 0.2% MCP (*w/v*). Different superscripts represent statistically significantly different (*p* < 0.05).

**Figure 3 foods-12-01068-f003:**
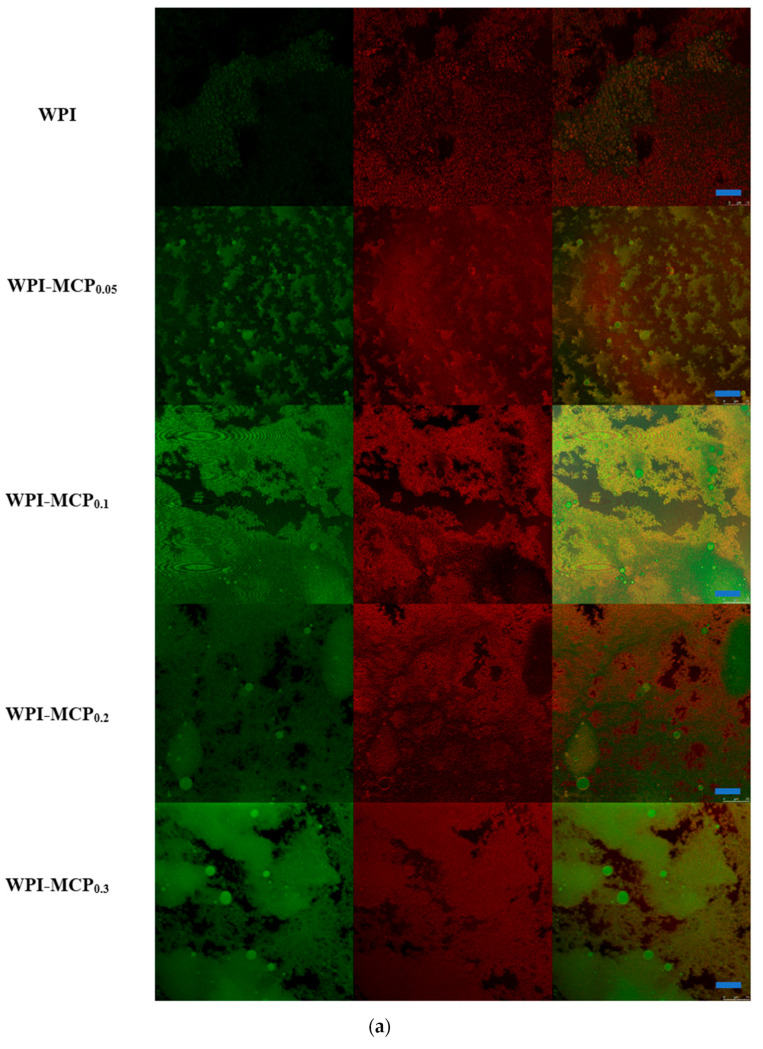
Typical CLSM (**a**) and light microscope images (**b**) of the emulsions stabilized by WPI (2%, *w*/*v*) with different concentration of MCP from 0 to 0.3% (*w*/*v*). For (**a**), from left to right were corn oil staining with Nile red, protein staining with Nile blue, and the combined images, respectively. The blue bars in (**a**) and black bars in (**b**) represent 25 μm and 20 μm in scale. The red arrows mark emulsion droplets.

**Figure 4 foods-12-01068-f004:**
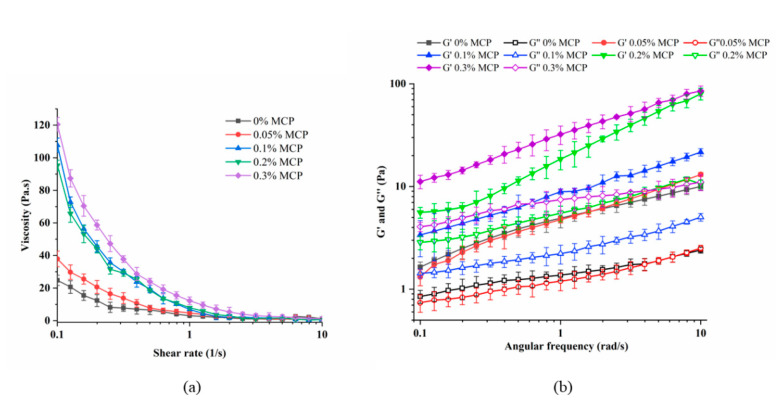
Apparent viscosity (**a**) and dynamic rheological properties (**b**) of WPI, WPI-MCP emulsions. Error bars are ± SD of the means.

**Figure 5 foods-12-01068-f005:**
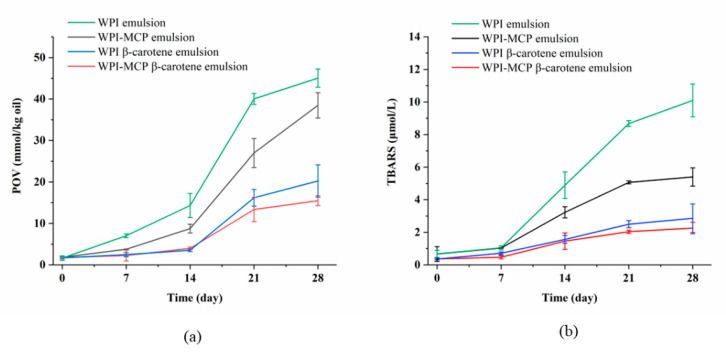
Changes of POV (**a**) and TBARS values (**b**) of different emulsions during 28 days storage.

**Table 1 foods-12-01068-t001:** The surface hydrophobicity of WPI, MCP and WPI-MCP conjugates.

Sample	Surface Hydrophobicity	R^2^
WPI	29.57 ± 0.93 ^e^	0.9925
MCP	3.07 ± 0.03 ^a^	0.9879
WPI-MCP_0.05_	20.60 ± 0.11 ^d^	0.9914
WPI-MCP_0.1_	11.68 ± 0.70 ^c^	0.9867
WPI-MCP_0.2_	9.07 ± 1.05 ^b^	0.9706
WPI-MCP_0.3_	8.53 ± 0.20 ^b^	0.9890

Note: The data are presented as means ± SD. Different superscripts in same column (a–e) represent statistically significantly different (*p* < 0.05).

**Table 2 foods-12-01068-t002:** The size, PDI, zeta potential and EE of β-carotene emulsions.

Emulsifiers	Size (nm)	PDI	Zeta Potential (mV)	EE (%)
WPI	175.5 ± 2.79 ^a^	0.29 ± 0.02 ^a^	−32.93 ± 1.05 ^b^	86.68 ± 2.45 ^a^
WPI-MCP	235.0 ± 2.03 ^b^	0.29 ± 0.04 ^a^	−37.87 ± 0.95 ^a^	87.18 ± 0.67 ^a^

Note: The data are presented as means ± SD. Different superscripts (a–b) in same column represent statistically significantly different (*p* < 0.05).

**Table 3 foods-12-01068-t003:** β-carotene retention rate of emulsions during storage.

Systems		β-carotene Retention Rate at Different Storage Times (%)				
		0 Day	7 Day	14 Day	21 Day	28 Day
WPI β-carotene emulsion	stored at 4 °C	100 ^a^	97.36 ± 0.3 ^a^	90.73 ± 11.98 ^ab^	89.13 ± 12.53 ^ab^	78.7 ± 14.03 ^ab^
	stored at 25 °C	100 ^a^	98.53 ± 0.51 ^b^	88.1 ± 1.75 ^a^	80.93 ± 3.4 ^a^	71.8 ± 4.42 ^a^
WPI-MCP β-carotene emulsion	stored at 4 °C	100 ^a^	97.1 ± 0.2 ^a^	96.3 ± 3.96 ^b^	98.03 ± 0.06 ^b^	86.33 ± 6.44 ^b^
	stored at 25 °C	100 ^a^	98.07 ± 0.75 ^ab^	99 ± 6.05 ^b^	78.8 ± 1.8 ^a^	73.57 ± 3.46 ^a^

Note: The data are presented as means ± SD. Different superscripts (a–b) in same column represent statistically significantly different (*p* < 0.05).

## Data Availability

Data is contained within the article.

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
