# Peer review of "Whey Protein Isolate-Mesona chinensis Polysaccharide Conjugate: Characterization and Its Applications in O/W Emulsions"

_foods, 2023, doi:10.3390/foods12051068_

Round 1

Reviewer 1 Report (Previous Reviewer 1)

Comments on “Characterization of Mesona chinensis polysaccharide-whey protein isolate conjugates and their application in stabilizing β-carotene emulsion”. Overall, it is the same content with the previous submission. 

1.       Please provide the picture of Mesona chinensis in the Introduction.

2.       Line 81-82. Corn oil was obtained from the local market. Please provide the location.

3.       Line 95. Equipment for freeze drying is needed.

4.       Line 148. Please provide the concentration of b-carotene in corn oil and final concentration of  b-carotene in emulsion?

5.       Line 167. It is better to use “Oxidative stability assessment”. Since the POV and TBARS are the indices of lipid oxidation.

6.       Figure 2a. Please indicate the peaks of the important functional groups.

7.       Figure 2b. Please provide the letters on the bars to indicate the statistical different.

8.       For all acronyms used in any Figure and Table, kindly supply their entire names. For figures/tables to be clear, they must stand on their own.

9.       Table 2. Please provide the letters to indicate the statistical different.

10.   Figure 5. Were the emulsions still safe to consume based on the POV and TBARS values? Were they rancid? Please specify the emulsion's permissible limit value for POV and TBARS. Also, based on these indices, please provide the shelf-life of the emulsion and suggest the strategies to prevent such changes.

11.   Oxidation or degradation products of b-carotene should be analyzed and make a discussion.

Author Response

Reviewer 2 Report (New Reviewer)

Brief summary:

The article is of interest and contributes significantly to the research in the area of developing conjugates and measuring the stability and bioavailability of encapsulated compounds by using pickering emulsions.

Title: Title is too long and should be made concise.

Line# 77: A diagram depicting the overall plan of study, analyses performed at various stages, etc at the beginning of materials and methods section, would provide readers with better clarity. Hence, strongly recommend inclusion of a diagram detailing, the overall experimental plan.

Line# 79: This article is based on the premise that the extracted material (MCP) is a pure polysaccharide. Hence a relevant concern related to purity of MCP polysaccharide needs to be addressed.  Cited authors’ own reference 22 is a study on Cyclocarya paliurus, while this study involves Mesona chinensis. One cannot conclude that the extraction process followed for one plant material provides same level of purity of extracted polysaccharide from other plant materials without confirmation.

Lines# 196- 209: The explanation in section 3.1 is lacking and needs re-writing. For example, Lines 202- 204 indicates there are changes in intensity of absorption peaks (1000-1500 cm-1) due to conjugate formation. However, in Figure 2a there is no spectra for MCP without conjugation. It is known that the changes in spectra are due to changes in functional groups. Hence a spectra for MCP is required to show the differences in peaks between MCP (without WPI) and WPI by itself. Spectra for the various WPI-MCP mixes are not clear to identify due to B&W figure. Hence determination of changes to heights of peaks with increasing MCP concentration in WPI-MCP mix is difficult.

The explanation in section 3.1 relates to MCP-WPI composite, while Figures 2a and 2b have legends showing WPI-MCP. The abbreviations used should be kept consistent throughout the manuscript.

Figure 2b: Were the difference shown in bar chart statistically significant? The figure needs to be updated with statistical information.

Lines# 228- 230: Surface hydrophobicity of MCP control sample (without WPI) would probably help clarify.

Lines# 240- 246: How do the authors explain the decrease in protein solubility between S2 and S3? In the method that the authors have used in this study, a sequential process of using the precipitate from S1 to prepare S2 and so on for S3 and S4 was not employed. Hence the comments by authors about increase of hydrophobic, hydrogen and disulfide bonds with the addition of MCP cannot be clearly established.

Figure 3a and 3b: Figures are not clear as they are B&W and hard to identify the features.

Figure 4a and 4b: Were these differences determined to be statistically significant?

Lines# 318-322 and Lines# 332-337: Error bars in figures 5a and 5b are overlapping. Are the differences statistically significant between encapsulated B-carotene emulsions containing only WPI or WPI-MCP?

Lines# 312- 336: The terms used in text are not consistent with the legends in figures. The entire manuscript should be reviewed to maintain consistency in terminology to maintain readability.

Line# 350: What were the standard deviations for the values 71.8%, 73.6%, 78.7% and 86.3% of B-carotene measured?

Figure#6: A colour figure would help in ease of identifying the treatments. Moreover, the error bars are indicative of large differences in values which overlap. Were the differences statistically significant?

Section 4 conclusions- This section may need re-writing based on above comments provided.

Author Response

This manuscript is a resubmission of an earlier submission. The following is a list of the peer review reports and author responses from that submission.

Round 1

Reviewer 1 Report

1.       Please provide the picture of Mesona chinensis in the Introduction.

2.       Line 81-82. Corn oil was obtained from the local market. Please provide the location.

3.       Line 95. Equipment for freeze drying is needed.

4.       Line 148. Please provide the concentration of b-carotene in corn oil and final concentration of  b-carotene in emulsion?

5.       Line 167. It is better to use “Oxidative stability assessment”. Sine the POV and TBARS are the indices of lipid oxidation.

6.       Figure 2a. Please indicate the peaks of the important functional groups.

7.       Figure 2b. Please provide the letters on the bars to indicate the statistical different.

8.       For all acronyms used in any Figure and Table, kindly supply their entire names. For figures/tables to be clear, they must stand on their own.

9.       Table 2. Please provide the letters to indicate the statistical different.

10.   Figure 5. Were the emulsions still safe to consume based on the POV and TBARS values? Were they rancid? Please specify the emulsion's permissible limit value for POV and TBARS. Also, based on these indices, please provide the shelf-life of the emulsion and suggest the strategies to prevent such changes.

11.   Oxidation or degradation products of b-carotene should be analyzed and make a discussion.